# Enhancing Resilience and Self-Centering of Existing RC Coupled and Single Shear Walls Using EB-FRP: State-of-the-Art Review and Research Needs

Ali Abbaszadeh ⬦ and Omar Chaallal *⬦

Department of Construction Engineering, École de Technologie Supérieure, University of Quebec, Montreal, QC H3C 1K3, Canada
* Correspondence: omar.chaallal@etsmtl.ca

**Abstract:** The primary seismic force-resisting system (SFRS) in middle- to high-rise reinforced concrete (RC) building structures often includes coupled shear walls (CSWs) and single shear walls (SSWs). These walls are designed to transfer lateral forces to the foundation and dissipate energy through the development of plastic hinges. The latter lead to residual displacement in these structural components. On the other hand, self-centering systems enable the structures to return to their initial position after severe loading or at least reduce residual displacement. The objectives of this study were, therefore, as follows: (i) to review the state of the art on shear wall self-centering techniques and retrofitting methods based on externally bonded fiber-reinforced polymer (EB-FRP); (ii) to evaluate research needs to improve the self-centering ability of shear walls using EB-FRP.

**Keywords:** residual displacement; self-centering; coupled shear wall; single shear wall; plastic hinge; FRP composite

## 1. Introduction

Most RC structures in medium- to high-rise buildings located in seismic and windy zones are made of single and coupled shear walls as lateral force-resisting systems. In this regard, it is crucial to find a way to deal with potential hazards and structural damage due to earthquakes, severe wind, and other potential lateral forces. The occurrence of severe earthquakes, even in buildings designed according to the most up-to-date modern building codes and standards, can cause structural elements to enter a nonlinear state and, as a result, experience residual displacement after the events, which can lead to their replacement in severe cases. Therefore, methods to increase the resilience of existing structures and strengthen their weaknesses are essential.

Self-centering systems enable structures experiencing residual displacement to return to their original and upright position after cyclic loading or reduce the residual displacement to the maximum acceptable by the standards, thereby enabling the structure to survive. In recent years, much research has been carried out to develop self-centering systems, although the use of such systems cannot be attributed to recent years or even decades because their traces can be seen even in historical buildings left by ancient civilizations. Nevertheless, the ultimate purpose of this study is to use these systems for existing structures.

In the meantime, the application of EB-FRP sheets for strengthening existing RC structures has been well-received by researchers in recent decades. Compared to other methods, using FRP sheets offers many advantages such as easy installation, high tensile strength, affordability, high resilience in different weather conditions, and the possibility of use on most structural elements. Moreover, unlike cement and steel, FRP sheets cause less environmental damage because their production generates fewer greenhouse gases. However, FRP laminates have some disadvantages, including lower modulus of elasticity

compared to steel, poor long-term dynamic performance (mainly due to debonding of the concrete medium from FRP sheets), and inadequate fire and heat resistance.

For a better understanding of and approach to the goals of this study, the main sections of this paper are organized as follows:

(i)     role and performance of shear walls in RC buildings;
(ii)    evaluation of strengthening existing RC structures using EB-FRP sheets;
(iii)   review of self-centering methods and their performance in RC buildings;
(iv)    conclusion and evaluation of required studies.

Figure 1 shows an overview of this research study.

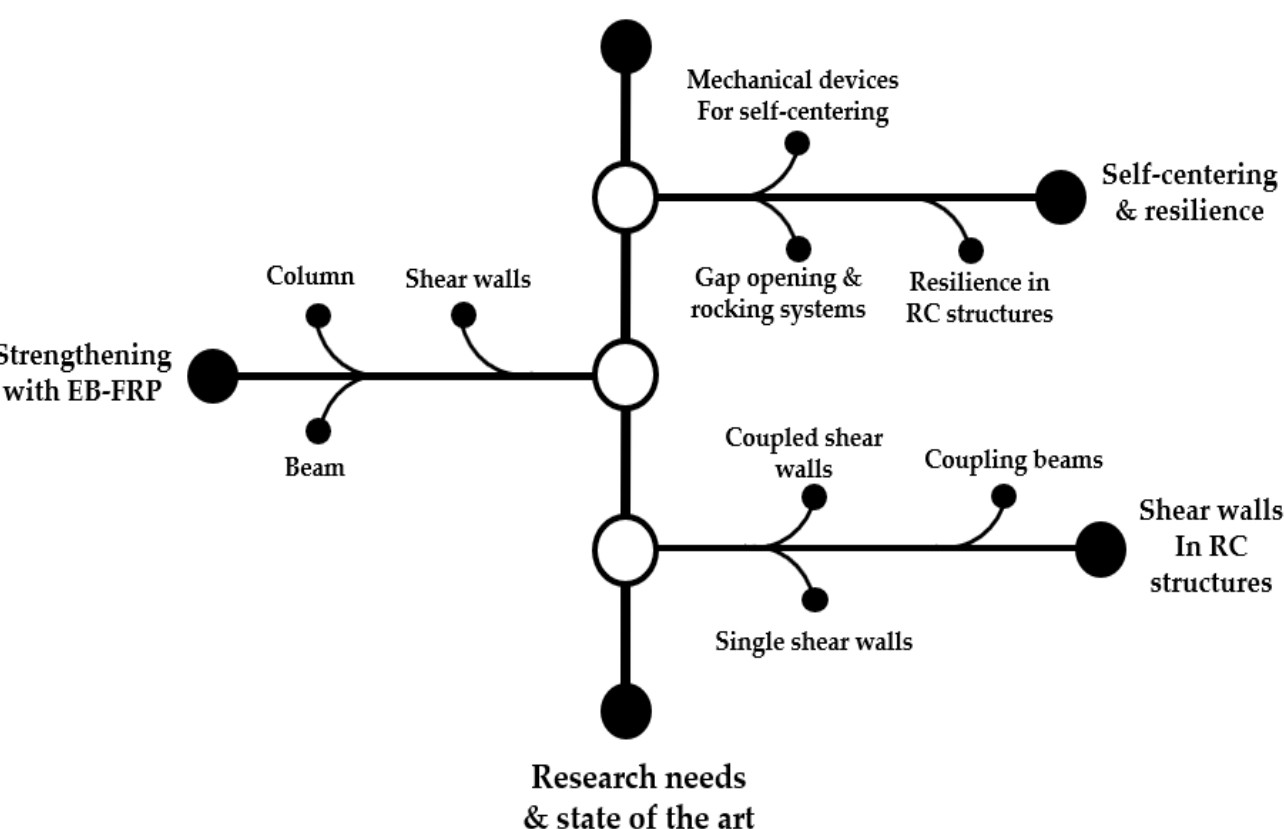

**Figure 1.** Overview of the present research study.

## 2. Shear Walls in RC Structures

Shear walls play a crucial role in a structure's resistance to lateral loads, gravity loads, and severe seismic forces. They prevent structural and nonstructural damage by reducing overall displacement and drift in the structure. Two shear wall configurations are often used in reinforced concrete buildings as lateral load-bearing elements: (i) single shear walls and (ii) coupled shear walls. The behavior of a slender single shear wall (SSW) can be compared to that of a reinforced concrete cantilever beam. These walls can be considered as upright cantilevers which tolerate tremendous amounts of bending moment, shear force, and compressive force due to lateral and gravity loads [1]. Accordingly, the moment–axial force interaction should be used to determine the strength of the critical section of a shear wall. When assessing the flexural capacity of a shear wall, vertical or flexural reinforcement in the web zone should be considered. As shown in Figure 2, when a lateral force is applied to single shear walls, they resist lateral forces by forming moment and shear at the wall base because the walls are typically considered fixed at the base.

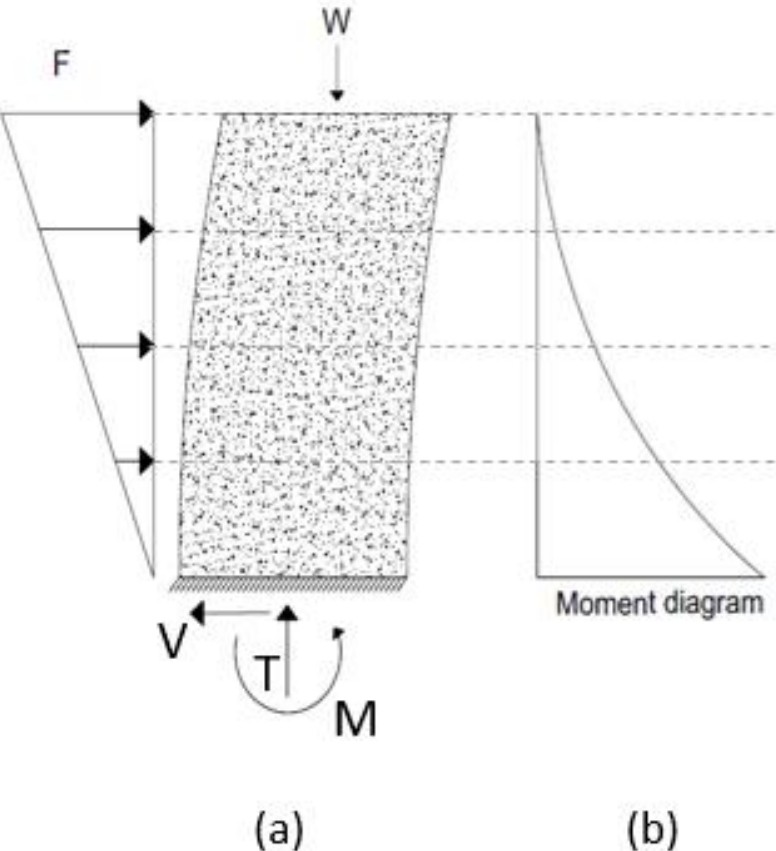

**Figure 2.** (**a**) Wall base reactions, (**b**) bending moment diagram of fixed-base SSWs exposed to lateral force.

In the early years, much research was conducted on the amount and configuration of vertical and horizontal reinforcement, wall thickness, concrete strength, and, more recently, the effect of the location of shear walls on their performance and the interaction of the frame and shear walls in structures. Later, the capacity design method for shear walls was developed [1]. A detailed overview of design requirements and a study on shear walls are available [2]. The importance and key role of the size and arrangement of vertical rebars on the ultimate curvature of shear walls were stated [3]. Bertero and Felippa [4] studied the effect of transverse confinement on buckling and ductility capacity. More recently, further research was carried out on the ductility of shear walls [5,6]. The occurrence of severe earthquakes and the observation of undesirable or unexpected failures in shear walls led to further developments in the design and strengthening of shear walls against seismic loads. In addition, some researchers investigated the effect of a shear wall's thickness on its performance during a severe earthquake [7,8].

Although single shear walls can perform properly against seismic forces, coupled shear walls (CSW) can be even more efficient due to their coupling system. These walls present many advantageous features such as high lateral stiffness and strength, superior architectural compatibility, and excellent energy dissipation capacity. Generally, CSW systems consist of two walls linked by coupling beams (CB). Fixed joints between the coupling beams and the walls that resist against moments can create an efficient rigid lateral force-resisting system. On the other hand, once a lateral force is applied to a CSW, in addition to the moment and shear force development at the wall piers base corresponding to shear force formation in the CBs, an axial tension–compression force couple is formed in each wall. This coupling system significantly improves CSWs' lateral resistance. Figure 3 shows force development in CSWs.

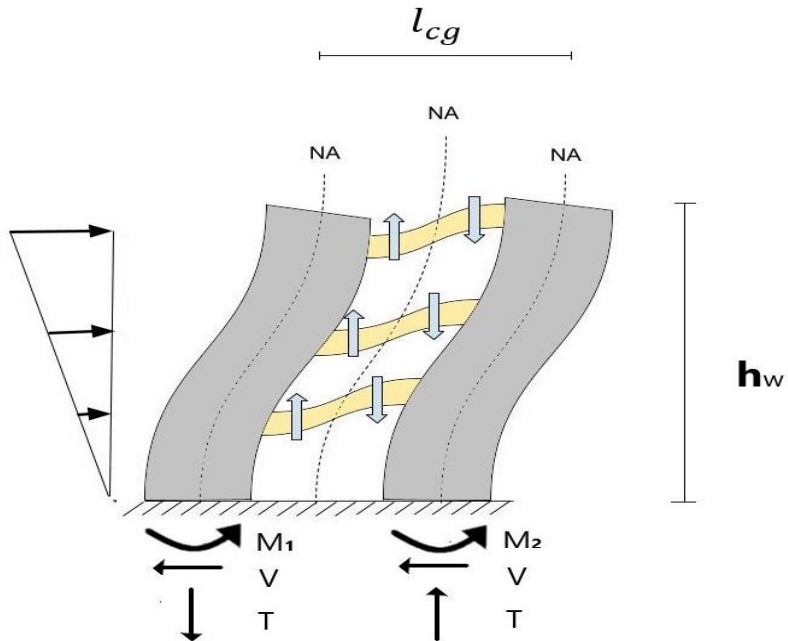

**Figure 3.** Force development in a CSW system.

Many successful studies have been conducted on CSWs, leading to tremendous development in their design and detailing [1,9,10]. These studies have shown that the stiffness and geometrical properties of CBs play a crucial role in the performance of these walls during a seismic event. To measure these features in CBs, the National Building Code of Canada (NBCC2015) [11] has introduced a ratio known as the degree of coupling (DC) as follows:

$$DC = \frac{Pl_{cg}}{M_1 + M_2 + Pl_{cg}} \tag{1}$$

where $P$ and $l_{cg}$ represent the value of the tension or compression force arising from the coupling function (in CBs) and the distance between the wall centroids, respectively, and $M_1$ and $M_2$ denote the internal moments generated in each wall pier. In addition, Chaallal et al. [12] proposed the following formula for the DC based on the statistical regression correlations between the DC and geometric features of CSWs:

$$DC = k\frac{H_b^a}{D_w^b + L_b^c} \tag{2}$$

where $H_b$ and $L_b$ are the coupling beam height and length, respectively, and $D_w$ is the wall length. Other dimensionless values, including $k$, $a$, $b$, and $c$, can be obtained from Table 1.

**Table 1.** Values of $k$, $a$, $b$, and $c$ for Equation (2) (adapted with permission from [12]. 1996, American society of civil engineers).

| Number of Stories ($n$) (1) | $k$ (2) | $a$ (3) | $b$ (4) | $c$ (5) |
|---|---|---|---|---|
| 6 | 2.976 | 0.706 | 0.615 | 0.698 |
| 10 | 2.342 | 0.512 | 0.462 | 0.509 |
| 15 | 1.697 | 0.352 | 0.345 | 0.279 |
| 20 | 1.463 | 0.265 | 0.281 | 0.190 |
| 30 | 1.295 | 0.193 | 0.223 | 0.106 |
| 40 | 1.190 | 0.145 | 0.188 | 0.059 |

In Equation (2), the author considered $\Upsilon$ equal to 0.6 as a coefficient to calculate the moment of inertia ($I_e$) of the walls and CBs ($I_e = \Upsilon I_g$, in which $I_g$ is the uncracked gross section). For other values of $\Upsilon$, including $\Upsilon_b$ (for the coupling beams) and $\Upsilon_w$ (for the walls), the DC can be determined using the following equation [12]:

$$DC = \frac{(\Upsilon_b/0.6)^{a/3}}{(\Upsilon_w/0.6)^{b/3}} \, k \, \frac{H_b^a}{D_w^b L_b^c} \tag{3}$$

As an appropriate SFRS, shear walls include many structural considerations and design requirements. Accordingly, in the equivalent static method, the base shear is significantly related to the type of the chosen SFRS. In this regard, (NBCC2015) [11] specified some criteria to determine the force at each floor through the distribution of base shear. Equations (4)–(6) calculate the base shear and its maximum and minimum limits:

$$V_{base} = S(T_a) M_v \frac{I_E}{R_d R_o} W \tag{4}$$

$$V_{max} = \max\left(\frac{2}{3}\frac{S(0.2)M_v I_E W}{R_d R_o}, \frac{S(0.5)M_v I_E}{R_d R_o}\right), \tag{5}$$

$$V_{min} = S(4.0) M_v \frac{I_E}{R_d R_o} W, \tag{6}$$

where $W$ denotes the total weight of the structure, which encompasses 25% of the snow load (SL) added to the deadload ($W = DL + 0.25\,SL$); $S(T_a)$ represent the design spectral response acceleration at the fundamental period; $M_v$ represents the factor considering higher-mode effects; $I_E$ represents the importance factor; $R_d$ denotes the ductility-related factor; and $R_o$ represents the overstrength-related factor. These values are related to the type of an SFRS. Then, the force on each floor should be calculated as follows:

$$F_i = (V_{base} - F_t)\frac{W_i h_i}{\sum_{i=1}^{n} W_i h_i} \tag{7}$$

where $F_i$ denotes the force at level $i$, $W_i$ denotes the weight assigned to the $i$th story, $h_i$ is the height of the $i$th story above the base, and $F_t$ is a part of the base shear that should be applied to the highest floor if the period of the structure is greater than 0.7 s to consider the effect of higher modes. $F_t$ is calculated as follows:

$$F_t = 0.07 T_a V_{base} \le 0.25 V_{base} \tag{8}$$

In addition, CSA/A23.3-14 [13] specified some requirements to ensure SSW ductility within and above the plastic hinge zone. In this regard, within the plastic hinge zone of shear walls, the plastic rotational demand ($\theta_{id}$) should be less than the plastic rotational capacity ($\theta_{ic}$). Equations (9) and (10) calculate $\theta_{id}$ and $\theta_{ic}$, respectively:

$$\theta_{id} = \frac{(\Delta_f R_o R_d - \Delta_f \gamma_w)}{(h_w - \ell_w/2)} \ge 0.004, \tag{9}$$

$$\theta_{ic} = \left(\frac{\varepsilon_{cu}\ell_w}{2c} - 0.002\right), \tag{10}$$

where $\Delta_f R_o R_d$ denotes the design displacement, $\Delta_f \gamma_w$ is the elastic portion of the displacement, and c denotes the neutral axis distance (an outermost compression fiber's distance from the neutral axis) calculated as follows:

$$c = \frac{P_s + P_n + P_{ns} - \alpha_1 \phi_c f_c' A_f}{\alpha_1 \beta_1 \phi_c f_c' b_w}, \tag{11}$$

where $P_s$ denotes the axial load at the section calculated as the sum of factored deadload, factored live load, and earthquake-load factors; $P_n$ denotes the earthquake-induced transfer force calculated from the interaction between CSW elements; $P_{ns}$ denotes the nominal net load applied to the segment due to yielding in the compression–tension of both concentrated and distributed rebars throughout plastic hinge development; $\alpha_1$ represents the ratio of middling stress within the compression block to the determined concrete strength; $\beta_1$ denotes the ratio of compression block depth to neutral axis depth; $b_w$ represents the wall thickness; and $A_f$ denotes the area of the flange; $\varepsilon_{cu}$ should be assumed as 0.0035 except for the compression zone of the wall containing special confinement reinforcement, according to CSA/A23.3-14 [13] requirements. For CSWs and CBs, $\theta_{id}$ should be calculated using Equations (12) and (13), respectively:

$$\theta_{id} = \frac{\Delta_f R_o R_d}{h_w} \tag{12}$$

$$\theta_{id} = \left( \frac{\Delta_f R_o R_d}{h_w} \right) \frac{\ell_{cg}}{\ell_u}, \tag{13}$$

where $h_w$ represents the height of the wall, $\ell_{cg}$ denotes the distance between the wall centroids, and $\ell_u$ represents the length of the clear span.

## 3. FRP Composites

All structures may be damaged or eroded due to various factors such as adverse weather conditions, traffic, severe earthquakes, changes in use, and other events. Hence, strengthening existing structures is inevitable because it is often more cost-effective than replacing the entire structure. Various methods have been investigated and proposed to strengthen concrete structures [14–16]. However, the use of FRPs to strengthen existing structures has gained tremendous popularity worldwide due to its ease of implementation and very high efficiency [17]. The first studies on utilizing FRP composites for the aerospace industry were conducted in the early 1960s [18]. Later, many numerical and experimental studies demonstrated the effectiveness of application of FRP composites to rehabilitate RC structures [19,20]. FRPs can be made of different fiber materials, including carbon, basalt, glass, and aramid fibers, which can be used to retrofit different components of structures [21,22]. They also feature many other advantages compared to conventional retrofitting methods, including high strength/weight ratio, anticorrosion properties, resistance to harsh weather conditions, enforceability in under-service intricate structures, and chemical resistance [23,24].

### 3.1. Strengthening RC Structures Using FRPs

As a strengthening method for RC structures, FRPs can be used in various forms and configurations, including externally bonded (EB) laminates, anchorage systems, and near-surface-mounted (NSM) bars [16,25]. EB-FRP is used in various ways to strengthen structures, including side bonding and partial or full wrapping as well as transverse and longitudinal strips on beams and slabs. Nevertheless, debonding of the concrete medium from FRP sheets is considered as a common drawback of EB-FRP. To overcome this problem, an anchorage system will help greatly to improve the performance of the strengthened structure [24,25]. FRP anchorage, spike anchorage, and mechanical fasteners are some of the anchorage systems used in different strengthening techniques. In addition to the anchorage challenge, FRP composites have some disadvantages, including (i) poor fire and heat resistance; (ii) elastic response up to rupture; (iii) vulnerability during drilling for anchoring; (iv) higher costs compared with traditional strengthening methods [21].

EB-FRP composites can be used to retrofit and repair a wide range of structural deficiencies; however, the type and condition of loading as well as the structural weakness must be carefully established before selecting the appropriate FRP strengthening configuration. Various deficiencies in columns, beams, and shear walls are described in the

following subsections, and an overview of different FRP strengthening patterns is shown in Figure 4. Siddika et al. [21] reviewed RC structure strengthening techniques using FRP composites comprehensively.

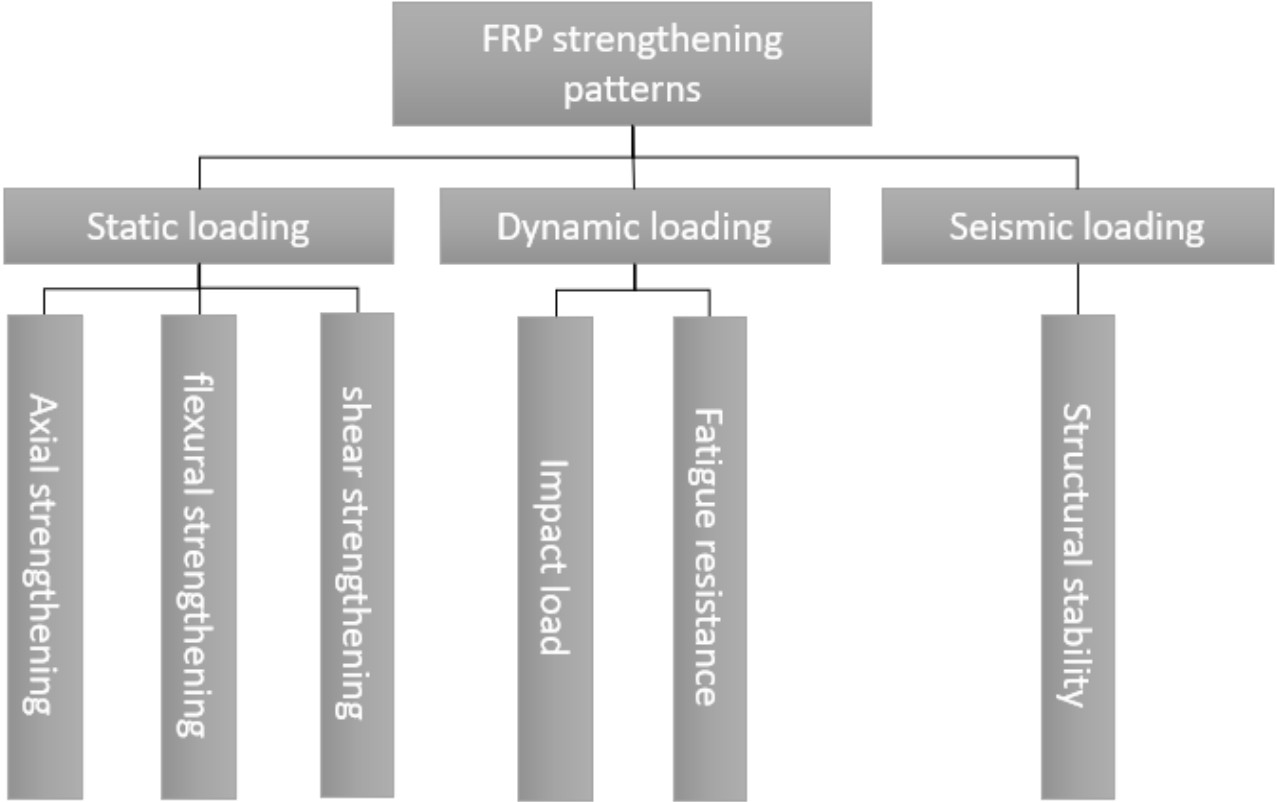

**Figure 4.** FRP strengthening patterns.

3.1.1. Columns

RC columns might be strengthened due to deficiencies such as (i) poor ductility, compressive, shear, and flexural strength [26–28]; (ii) lack of lap splice and confinement [29–31]; (iii) poor column–beam connection [32,33].

One of the most prevalent methods of increasing compressive strength in RC structures, especially in columns, is the use of fully wrapped EB-FRP around the perimeter of the element. This can significantly enhance the confinement (lateral pressure) on the column, thereby improving ductility, shear strength, axial strain, and load-carrying capacity [21,34]. Note that confinement by EB-FRP is more effective in circular columns than in rectangular ones due to the larger corner radius of the latter [18]. Studies have also shown that the effectiveness of confinement using the EB-FRP wrap on circular columns differs from that on rectangular sections. Although axial strength in circular columns is the most influenced by confinement, in rectangular-section columns, confinement primarily enhances concrete strain capacity [35,36]. Razavi et al. [37] conducted tests under axial eccentric cyclic loading in six RC columns: non-strengthened, strengthened with longitudinal EB-FRP sheets, and strengthened with fully wrapped FRP sheets. The results showed that using longitudinal EB-FRP sheets, despite the decrease in ductility, can increase the bearing capacity under high eccentric loads by up to 60%. In contrast, fully wrapped FRPs do not improve the bearing capacity, but significantly increase ductility. Yi et al. [38] showed that longitudinal and transverse EB-FRP sheets are required for RC columns under eccentric loads to improve ultimate strength and ductility respectively. More studies have been carried out on the behavior of FRP-strengthened RC structures under eccentric axial loads [39–42]. Moreover, an applied concentric cyclic axial load has been considered by many researchers [43–45].

Poor seismic design can lead to shear brittle failure before flexural failure in columns [46]. Moreover, shear failure is more common in short columns. However, EB-FRP can increase the shear strength of columns and alter shear failure to flexural failure. Using a finite element model [47] revealed that an EB-FRP wrap could improve the shear strength of a damaged column under impact load and alter the failure mode from brittle shear to ductile flexural. Mo et al. [48] revealed that using FRP sheets considerably increased the ductility and shear capacity of hollow rectangular bridge columns. Iacobucci et al. [49] used an FRP jacketing technique to strengthen columns with inadequate seismic design and detailing. The results showed a considerable improvement in energy dissipation and ductility. Kargaran et al. [50] conducted finite element analysis and experimental tests on six short RC columns under cyclic lateral displacement to evaluate the effectiveness of the EB-FRP technique in different schemes such as transverse, diagonal, and hybrid. The results revealed that when these techniques were used, the failure mode in columns was altered from shear to flexural, and ductility and energy dissipation improved.

The EB-FRP wrap (jacketing) enhances the sectional curvature capacity of an RC column. Taking advantage of this feature and providing an adequate length of the EB-FRP wrap in the plastic hinge zone greatly increased the drift capacity of the RC column [51]. Hence, many studies have taken advantage of the FRP jacketing technique to improve RC column performance [52,53]. Parvin et al. [54] conducted a nonlinear finite element analysis on columns retrofitted in the plastic hinge zone using FRP jackets under axial and cyclic lateral loads. The results revealed that this technique considerably increased ductility and strength, leading to a significant delay in stiffness degradation of the RC column.

The importance of column–beam connections is as great as that of other structural components during seismic loading. Even if the beams and columns remain intact, the entire structure can collapse if the column–beam connections fail [55]. One of the most common failures in column–beam joints is shear brittle failure, which should be avoided through appropriate design and detailing or rehabilitation techniques in existing structures. FRP sheets can improve the flexural capacity of the connections and decrease joint rotation [56]. Many studies have been conducted to evaluate the effectiveness of FRP composites as a beam–column joint strengthening technique [57,58]. Ghobarah et al. [59] showed that using GFRP jacketing in column–beam connections can enhance energy dissipation, shear resistance, and ductility and change shear brittle failure to flexural failure in joints.

Many buildings experience total structural destruction after a seismic event, mainly due to column failure and shear and flexure deficiencies. Hence, to avoid these incidents, columns should be strengthened. Ouyang et al. [60] showed that fully wrapped EB-FRP in columns enhances axial strength and confinement and increases ductility and column failure control. In addition, it greatly enhances energy dissipation capability in columns. Iacobucci et al. [49] studied columns strengthened with the fully wrapped EB-FRP technique. The results revealed a 54% improvement in most parameter values and resistance against seismic forces. In addition, it has been shown that using hybrid EB-FRP and NSM techniques considerably increased ductility and load capacity during seismic loading [61]. Murad et al. [62] evaluated the effect of EB-FRP on the cyclic response of unconfined joints constructed with recycled concrete. The results showed great load-carrying capacity and strength enhancement after strengthening components with FRP. More detailed evaluations on seismic retrofitting of RC connections have been presented in recent studies [63,64].

### 3.1.2. Beams

EB-FRP strengthening of RC beams is a widespread method that has shown great effectiveness in practice [65,66]. The most common applications of EB-FRP sheets in RC beams are as follows: (i) shear strengthening [67–69]; (ii) flexural strengthening [70,71]; (iii) fatigue strengthening [72,73]; (iv) impact strengthening; (v) seismic strengthening.

All RC beams are subject to bending. Flexural failure mode happens due to exposure to loads that are higher than the flexural capacity of the beam. Depending on the

beam type and loading, FRP sheets or laminates with or without anchors can be used to strengthen these elements against bending. Applying unidirectional FRPs on the tension side of the element is the most common method of flexural strengthening [70,74,75]. Panahi et al. [66] evaluated the flexural retrofitting of RC beams with a combination of EB-FRP layers and near-surface mounted FRP rods. The results showed an acceptable increase in flexural capacity, stiffness, ultimate bending moment, and decrease in ductility. El Ghadioui et al. [76] investigated the flexural retrofitting of RB members under long-term cyclic loading. Zhang et al. [77] showed that using NSM strips can achieve a 200% gain in flexural resistance. Attari et al. [78] showed that using EB-FRP results in a decrease in deflection as well as in the length of bending cracks. Extensive studies have been conducted on the flexural behavior of RC beams strengthened with FRP sheets [79,80].

The application of FRP composites as an effective RC beam shear strengthening method has been evaluated in many studies, which have proven that the orientation of FRP sheets as a significant feature governs its efficiency [81]. Singh [82] showed that inclined EB-FRP wrapping is more efficient in enhancing shear capacity. The use of wrapping oriented at 45° could avoid diagonal cracks. Using a hybrid U-wrapped FRP and anchorage system, Baggio et al. [83] evaluated the repair of shear cracks caused by a prestressed force. They showed that the increase in load-carrying capacity could reach 82.2%. Numerous researchers also investigated shear strengthening using FRP sheets [84–86].

Impact load originates from many sources such as moving vehicle load, falling rocks or weights, and explosions. The most probable damage due to this loading pattern is shear failure. Hence, increasing an element's shear capacity is crucial to improving impact strength. Tang et al. [87] argued that EB-FRP on RC beams considerably benefits impact strength. Their study showed 15% and 96% gain for impact and shear strength, respectively. In this context, further research evaluated the impact resistance of RC members strengthened with FRPs [88,89]. Although shear strengthening is more common for impact resistance improvement in RC beams, many studies have shown that flexural strengthening significantly develops the impact resistance of this component [90,91]. Kishi et al. [92] revealed that flexural strengthening could improve the impact resistance of RC beams. They carried out an experimental evaluation of RC beams strengthened in flexure using both aramid FRP (AFRP) and carbon FRP (CFRP) sheets. The impact load was simulated by a steel weight that was released from different heights, and the increase in the dropping height continued until the debonding of FRP and concrete.

A beam exposed to cyclic loading gradually loses its ductility and stiffness, leading to permanent deformation. Aidoo et al. [93] showed that using FRPs with high tensile strength reduces stress concentration at the cracks in the elements and avoids fatigue in the structure. Alyousef et al. [94] showed that because the stress flow occurs in the FRP–concrete interface, fatigue control depends entirely on concrete, epoxy resin, and FRP strength. They also revealed that using U-wrapped EB-FRP improved shear capacity and rigidity and decreased crack length under repeated loadings, increasing overall fatigue performance by 62%. Charalambidi et al. [95] showed that using EB-FRP in RC beams to increase flexural strength could improve the fatigue resistance of internal steel rebars. Therefore, elastic materials can help maintain strength and stiffness against fatigue loading. Meanwhile, it should be considered that FRP–concrete beams may experience fatigue crack propagation due to cyclic loading. Further discussions have been held on the fatigue resistance of FRP-strengthened RC beams [96,97].

### 3.1.3. Shear Walls

Stiffness degradation in shear walls during cyclic loading may alter their efficiency and performance. These concerns have led to the development of various innovative strengthening methods for RC shear walls [98–100]. Ehsani et al. [101] showed that strengthening shear walls with FRPs is the most cost-effective technique. However, it can be argued that the functionality of a shear wall strengthened with FRPs depends on two major factors: the FRP configuration and the anchorage pattern between the FRP strips and the shear

wall [102–104]. Altin et al. [104] revealed that the configuration using lateral FRP strips featured the best performance among the various FRP configurations to improve lateral displacement and load-carrying capacity.

In another study by Honarparast et al. [105], two CBs with a span-to-depth ratio equal to two were experimentally evaluated. One was conventionally reinforced and detailed based on NBCC 1941, and the second was diagonally reinforced and detailed based on NBCC 2015 and CSA A23.3 14 [13]. The coupling beam dimensions were 500, 250, and 1000 mm in depth, width, and span, respectively. Four 20 mm diameter rebars were used for conventionally reinforced specimens as top and bottom longitudinal reinforcement. The development length, which is vital to calculating the tensile resistance of rebars in the walls, was determined according to the NBCC codes. For shear resistance, 10 mm diameter hooped rebars with 200 mm spacing were used. Six horizontal 10 mm diameter rebars and eight diagonal 22 mm diameter rebars were used for the diagonal reinforcement specimen. Five 10 mm diameter rebars with 100 mm spacing were used for shear resistance. Both CBs were submitted to reversed cyclic loading. The study revealed that the diagonally reinforced CB outperformed the conventional CB because the load resistance and the energy dissipation in the diagonally reinforced CB increased up to 4.4 and 10.2 times, respectively. It also showed that stiffness degradation was less in the diagonally reinforced CB (4%) than in the conventionally reinforced CB (43%). In addition, the diagonally reinforced CB featured significantly less pinching in hysteretic loops. According to the results, at around 138 kN load, the conventional specimen rebars started to yield, and shear cracks due to sliding shear failure emerged at the joint locations in the early stages of reversed cyclic loading. This contrasts with the diagonally reinforced CB, where rebars started yielding at a load of 495 kN. These results showed the need for seismic rehabilitation of old structures designed and detailed based on old codes and standards. In addition, Honarparast et al. [106] evaluated a new method for retrofitting CBs with EB-FRP. To that end, two conventionally designed RC specimens (according to NBCC 1941) were considered: one control (unrehabilitated) specimen and one specimen strengthened with EB-FRP. Both were subjected to cyclic loading. A comparison of experimental results revealed a significant improvement in energy dissipation and hysteretic behavior with less stiffness degradation. The results revealed that as the 138 kN load was approached, the rebars in the control specimen started to yield, and shear cracks due to sliding shear failure at the joint locations emerged in the early stages of reversed cyclic loading. For the specimen strengthened with EB-FRP, the load reached 308 kN at the end of reversed cyclic loading, and small cracks appeared near the joints. Debonding of FRP strips at the joints also appeared during the load cycles. The hysteretic behavior of the control specimen featured large pinching, stiffness degradation, and loss of the energy dissipation capacity. The load-carrying capacity suddenly dropped after the displacement reached 24 mm. In contrast, the rehabilitated specimen did not show any considerable pinching and stiffness degradation before FRP debonding, resulting in a rapid load reduction at the final load cycle. Several schemes for strengthening SSWs and CSWs with EB-FRP are presented in Figure 5.

The other critical issue in RC shear wall strengthening with EB-FRP is the premature debonding of FRP sheets, which inhibits the FRPs from reaching their strength capacity. As mentioned earlier, epoxy resins are generally used as an adhesive to fix FRP sheets or strips to the concrete surface. However, due to severe cyclic loading, these resins lose their strength and lead to debonding. Hence, research activities are still ongoing to find the best solution to this problem. Studies have also revealed that hybrid application of U-shaped FRP-bonded sheets with a metal anchor is an appropriate technique to avoid premature debonding [107,108].

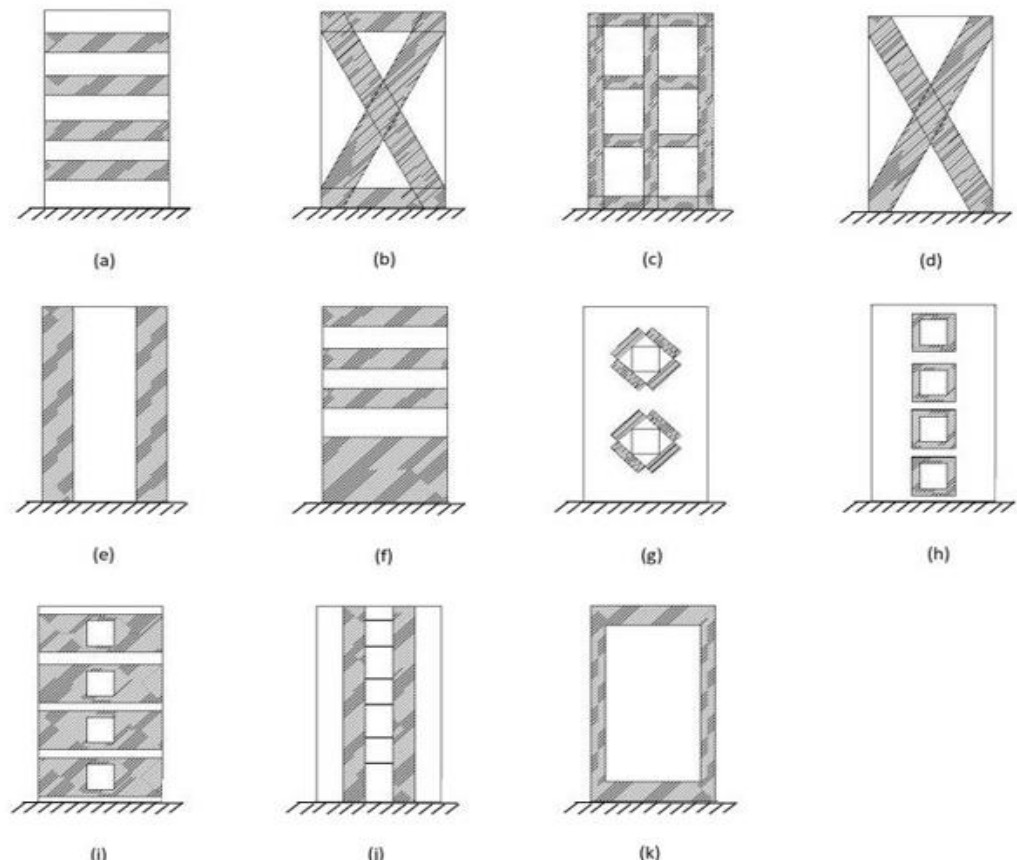

**Figure 5.** Typical schemes of strengthening shear walls with EB-FRP, where (**a**–**d**) modified from Sakr et al. [109]; (**e**) modified from Elnady [110]; (**f**) modified from Layssi et al. [111]; (**g**–**j**) modified from Aslani et al. [112]; (**k**) modified from Arabzadeh et al. [113].

Sakr et al. [109] showed that configuration (c) in the walls governed by shear failure has the most satisfactory performance in order to increase ductility and lateral strength. Nevertheless, pattern (b) has the best performance for shear walls governed by flexural failure. They also indicated that in diagonal configurations (a, b), bonding overall 50% (from each end) of the diagonal length of FRP sheets is as adequate as complete bonding of FRP sheets. Elnady [110] revealed pattern (e) can improve ductility and energy dissipation and avoid brittle failure in shear walls. Layssi et al. [111] showed that (i) scheme (f) avoids brittle and side-splitting lap splice failure by providing sufficient confinement in the plastic hinge region; (ii) one layer of fully wrapped FRP within the plastic hinge zone of shear walls can delay the buckling of flexural rebars and improve shear strength; (iii) this configuration can increase the energy dissipation of shear walls. Aslani [112] revealed that configurations (g, h, i, and j) could increase energy dissipation and bearing capacity and decrease residual displacement of shear walls. They also showed that configurations (g, h) are more effective in shear walls with central openings, but pattern (g) cannot be used for large-size openings. Arabzadeh et al. [113] showed pattern (k) could significantly increase ductility and energy dissipation of squat shear walls.

## 4. Self-Centering and Resilience

At high lateral forces, the primary role of shear walls in RC structures is to dissipate a significant amount of energy through plastic hinges that form within the base of the wall and in the CBs of CSWs, as well as through reinforcement yielding. On the other hand, formation of plastic hinges results in residual displacement in SSWs and CSWs (see Figure 6a). Self-centering systems enable structures experiencing residual displacement to return to their original upright positions after cyclic loading or at least reduce the residual

displacement to the maximum acceptable amount, thereby enabling the structure to survive. Typically, hysteretic behavior of an ideal self-centering system is a flag-like shape which features no residual displacement after a cyclic loading (see Figure 6b). The self-centering ability has been studied for many structural components, including columns, frames, walls, and bridge piers. Note that the application of this method is not limited to RC structures (which is the scope of this study), and several studies have been conducted to evaluate self-centering ability in timber, steel, and masonry structures. Various self-centering techniques can be used to restore a structure to its previous state. Two main techniques for creating self-centering in structures are rocking action and use of mechanical devices [114,115]. The following subsections are devoted to reviewing the application of these self-centering techniques in RC structures.

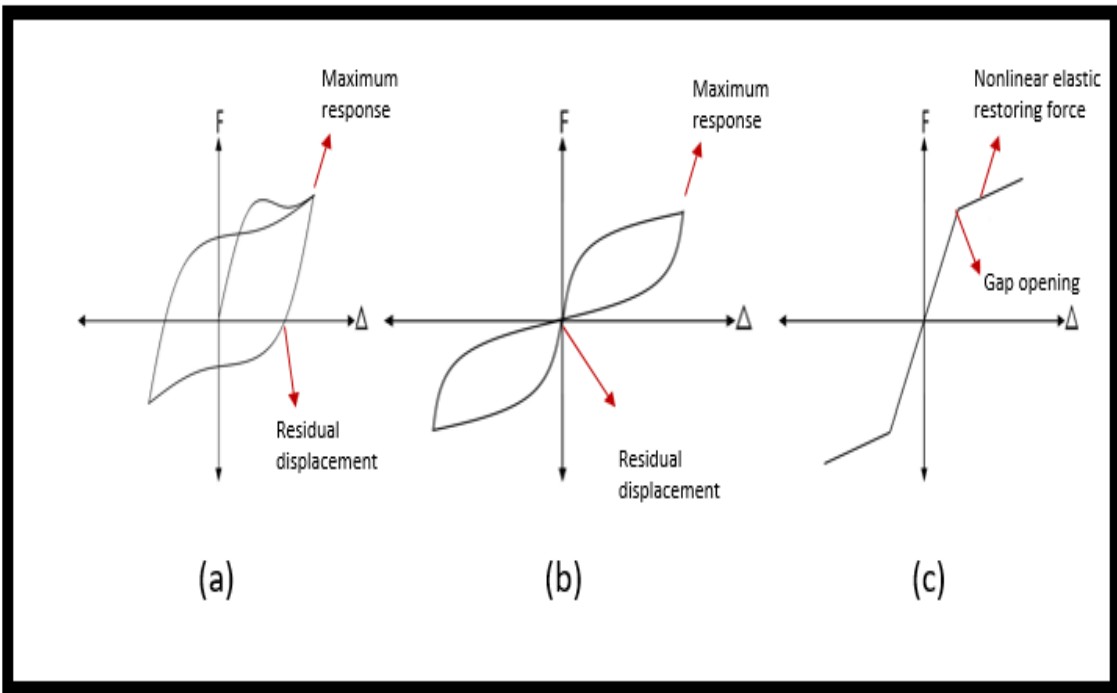

**Figure 6.** (**a**) Hysteretic behavior of conventional shear walls, (**b**) typical hysteretic behavior of ideal self-centering systems, and (**c**) restoring force in rocking systems and gap opening using PT tendons.

*4.1. Gap Opening and Rocking Systems*

In this method, uplifting the whole structure or rocking major structural components can mechanically resist seismic force and decrease damage and residual displacement. When this occurs, structural components remain elastic through a nonlinear softening response. Rocking systems may use either post-tensioned (PT) tendons (controlled rocking) or gain self-weight (uncontrolled rocking) [116]. In controlled rocking systems, structural components are pre-compressed at their surfaces using PT elements. Once the lateral force nullifies the precompression force, the joint section decompresses, and a gap opens. Thereafter, PT elements recover the lost stiffness of the lateral resistance system by forming axial stiffness. If the strain at the joint overcomes the elastic strain limit of the PT element, the system may fail [116] (see Figure 6c). In addition, some hybrid techniques in controlled rocking systems are equipped with dampers or restraining devices [117]. Numerous research studies on self-centering with unbonded post-tensioning and energy dissipators have also been conducted [118–120]. Holden et al. [121] showed that using a prestressed concrete system makes it possible to create the necessary force to return the structural element to its initial state. In this case, tendons should be unbonded over a certain length to remain in the elastic range during loading. This process creates the necessary force to deal with large displacement and prevents the formation of plastic hinges at the wall base and, consequently, residual displacement. Furthermore, because concrete is not bonded

to tendons and the lateral strength system does not rely on the interaction between rebars and concrete to withstand lateral loads, cracks in the concrete are significantly reduced. In addition, the minor damage that may occur in the bottom corners of the walls can be avoided by using foundation-embedded steel mating plates in these areas. Post-tensioning can be used for moment-resisting RC frames as well. This method is also based on a beam-and-column joint gap opening system [116]. In addition, energy dissipator devices located at the joints can play a significant role in absorbing seismic loading energy. Such systems can tolerate significant rotation with no loss in the load-carrying capacity [122]. Some achievements regarding the investigation of self-centering of moment-resistant RC frames have also been reported [122–125].

In uncontrolled systems, the self-weight of structural components such as beams, slabs, and columns causes moment resistance at the structure's base and thereby lateral resistance for the whole structure [126,127]. The energy dissipation mechanism in these systems occurs through sliding, impact, and friction between structural components. In the second method, structural components benefit from innovative devices to gain more self-centering and energy dissipation ability. The next subsection briefly discusses one of the most commonly used techniques in this field. Figure 7 shows various self-centering systems, which are comprehensively reviewed by Zhong et al. [114].

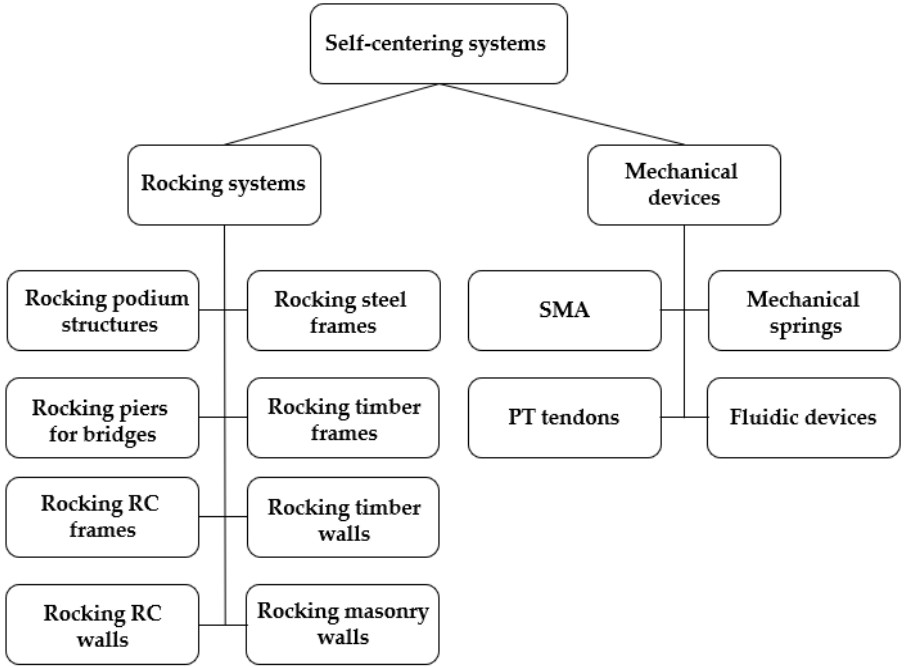

**Figure 7.** Various self-centering systems.

### 4.2. Mechanical Devices for Self-Centering

One of the most relevant techniques for self-centering is the use of shape memory alloy (SMA)-based systems. The super-elasticity property of SMA systems leads to the reduction of deformation after unloading. Indeed, SMA systems may recover up to 10% strain in some applications [128]. Therefore, these techniques are widely used in various industries such as medical, mechanical, aerospace, and civil engineering [129,130]. This ability, coupled with its inherent energy dissipation, makes SMA a good choice for self-centering systems. However, SMA wires may present some disadvantages related to their reduced load capacity and challenging anchorage due to their slippery surfaces. Hence, devices that work based on SMA wires are mostly designed for uniaxial tensile loading cases, which restricts their wide use [131]. Several research studies have been conducted to develop self-centering systems using the SMA technology [132,133]. Soares et al. [134] compared lateral strength, energy dissipation, drift capacity, and recentering ability of conventionally reinforced and SMA–steel-reinforced shear walls. Specimens for both reinforcement types

were considered as 10-story shear walls designed for both eastern and western Canadian seismic zones, with similar geometry and reinforcement configuration. All the specimens were evaluated through nonlinear pushover and reverse cyclic finite element analysis. The results showed similar lateral strength and drift capacity for both steel and hybrid SMA–steel-reinforced walls, but the recentering ability of the walls with hybrid SMA–steel reinforcement was significantly higher. However, walls with steel reinforcement had more energy dissipation capacity. Nevertheless, due to the drawbacks and shortcomings of SMA, in many studies, hybrid systems using energy dissipators such as hysteretic, viscous, and frictional dampers are considered [133,135,136].

Another innovative technique for enhancing self-centering in structures is the application of fluidic devices. These practical devices have the main characteristics of a damper and an internal pressurized fluid action that provide a recentering force in a structural component [137]. These devices' detailed application and design steps in structures were discussed by Kitayama et al. [138].

### 4.3. Resilience in RC Structures

Structural seismic resilience is the ability to undergo large lateral displacements with the minimum level of damage, plastic hinge formation, and residual deformation through inherent self-centering ability or attached self-centering systems. Seismic-resilient structures can dissipate a large amount of energy and rapidly return to their initial position with little permanent deformation, which helps maintain the serviceability of the structures after severe seismic events.

Conventional steel-reinforced shear walls have self-centering ability within elastic deformations. However, as these walls experience large plastic deformations followed by concrete crushing, reinforcement yielding, and residual displacement, they significantly lose their self-centering ability. In this regard, in recent decades, many studies have been carried out to increase the self-centering ability and resilience in shear walls.

In ancient buildings, structures benefited from uncontrolled rocking systems in wide columns. These structural components took advantage of their self-weight to form moment resistance and resist lateral forces [139]. Such systems are not as common as shear walls in modern buildings due to their poor seismic performance. In contrast, post-tensioned precast RC shear walls as an alternative SFRS showed high efficiency under seismic loads. This method is based on a gap opening system in which longitudinal PT elements supply a recentering force in precast shear walls after gap opening at the base of the walls. Many studies have illustrated the performance improvement of these lateral force-resisting systems compared to conventional precast RC shear walls [140–142]. Some studies have also investigated the use of energy dissipators incorporating PT elements [119,143,144]. This method has also been applied to CSWs, where gap opening formation was considered at the wall base and the wall–CB joints. Changes in the DC have been investigated, and improvements in ductility, energy dissipation, and strength have been observed [145,146].

As another efficient technique, using EB-FRP composites can improve resilience in structures where a large amount of seismic energy can be dissipated through reinforcement yielding and concrete crushing. In contrast, inherent elastic behavior and high strength of FRP composites can provide the required self-centering forces and help the structure return to its primary position.

Many studies have evaluated the use of FRP reinforcements to improve blast resilience of concrete structures [147,148]. Mutalib et al. [149] carried out experimental studies on RC panels strengthened with EB-FRP. The study was conducted on four walls of the same geometrical characteristics, including a non-strengthened wall ($w_1$), an FRP-strengthened wall with no anchor ($w_2$), an FRP-strengthened wall with anchors at the boundary ($w_3$), and an FRP-strengthened wall with distributed anchors ($w_4$). The results revealed that using EB-FRP can decrease the residual displacement of walls under blast loads. More studies have been carried out on applying FRP bars to achieve seismic-resilient RC structures. Billah et al. [150] evaluated the hybrid application of SMA and FRP bars in columns

to achieve an acceptable level of resilience in RC structures. The study showed an improvement in energy dissipation and a significant decrease in residual displacement. The application of FRP bars improved resilience in frames [151]. In this regard, applying EB-FRP sheets can be a beneficial choice for strengthening RC structures to achieve higher resilience and serviceability after seismic events [152]. Abbass et al. [153] studied the hybrid application of EB-FRP sheets and SMA to decrease the residual displacement of bridge columns. Although many studies have been conducted on applying FRP bars and SMA to achieve more resilient structures, most of them are appropriate for designing new structures. In this regard, EB-FRP sheets can be a good choice for existing complex structures. However, few studies have been dedicated to evaluating this technique's advantages. Figure 8 shows an overview of the general self-centering systems in RC shear walls.

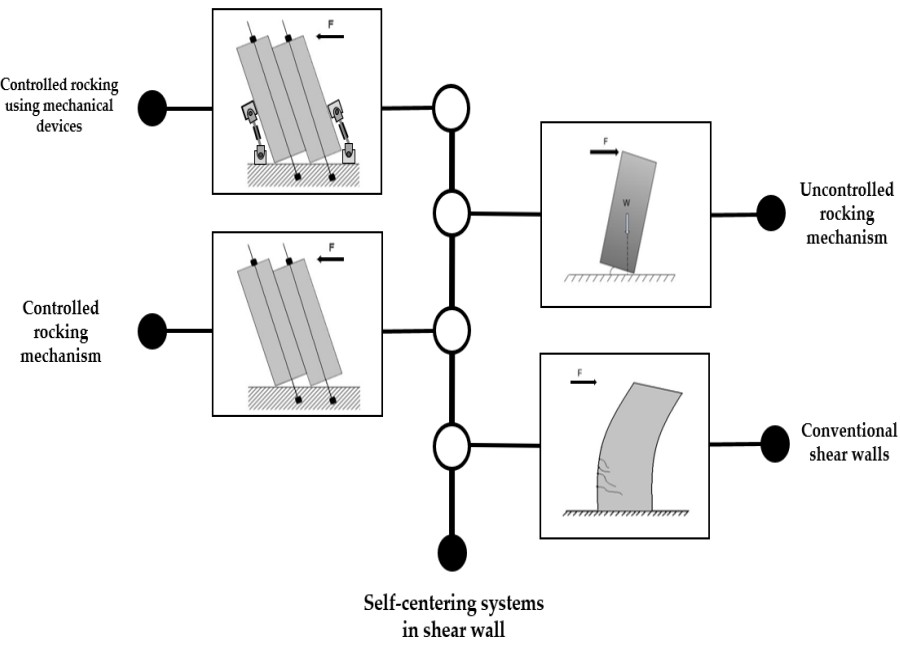

**Figure 8.** General self-centering systems in shear walls.

A summary of 74 research studies on the evaluation of different self-centering methods can be found in Table 2. The statistical results in this table can contribute to figuring out the areas in which less research has been done so far. The contributions of the various structural components and methods evaluated in these studies are summarized in Figures 9 and 10, respectively. The statistical analysis underlying Table 2 is used in Sections 5 and 6 to describe the research needs and conclusions in this study.

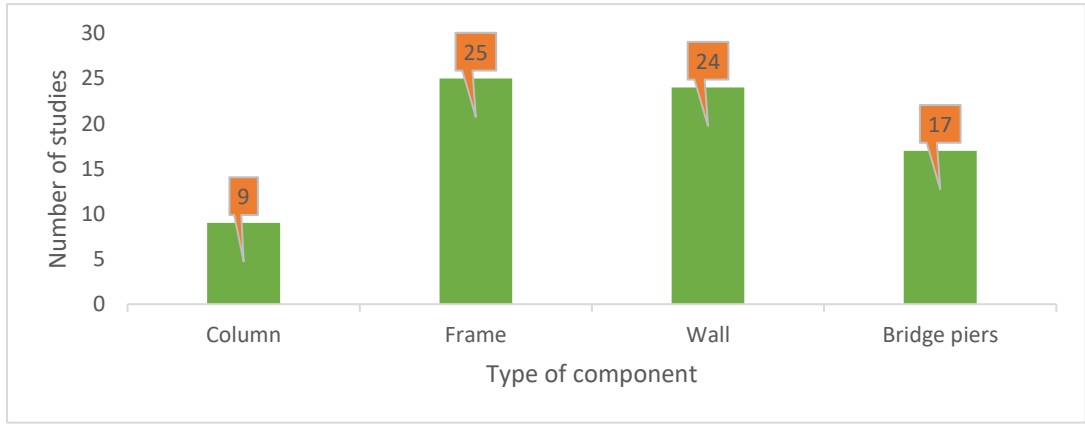

**Figure 9.** Visualization of the summarized research based on the investigated components.

**Table 2.** Summary of self-centering studies regarding method, structural components, and evaluated parameters. Note: a—rocking podium structures, b—rocking piers for bridges, c—rocking RC frames, d—rocking RC walls, e—rocking steel frames, f—rocking timber frames, g—rocking timber walls, h—rocking masonry walls, i—SMA, j—springs, k—PT tendons, l—fluidic devices, SR—self-centering capacity or residual displacement, ED—energy dissipation, SO—stability and overturning, St—stiffness, and DR—other dynamic responses.

| Study | Year | Method | | | | | | | | | | | | Structure | | | | Evaluated Parameter | | | | |
|---|---|---|---|---|---|---|---|---|---|---|---|---|---|---|---|---|---|---|---|---|---|---|
| | | a | b | c | d | e | f | g | h | i | j | k | l | Column | Frame | Wall | Bridge Piers | SR | ED | SO | St | DR |
| [154] | 2014 | ✔ | | | | | | | | | | | | ✔ | | | | ✔ | ✔ | | | |
| [155] | 2015 | ✔ | | | | | | | | | | | | ✔ | | | | ✔ | ✔ | | | |
| [156] | 2013 | ✔ | | | | | | | | | | | | ✔ | | | | | | ✔ | | |
| [157] | 2002 | ✔ | | | | | | | | | | | | ✔ | | | | ✔ | | | | |
| [158] | 2009 | ✔ | | | | | | | | | | | | ✔ | | | | | | ✔ | | |
| [159] | 2012 | ✔ | | | | | | | | | | | | ✔ | | | | | | ✔ | | |
| [160] | 2015 | ✔ | | | | | | | | | | | | ✔ | | | | | | | ✔ | |
| [161] | 2019 | ✔ | | | | | | | | | | | | ✔ | | | | | | | | ✔ |
| [162] | 2020 | ✔ | | | | | | | | | | | | ✔ | | | | ✔ | ✔ | | | |
| [163] | 1997 | | ✔ | | | | | | | | | | | | | | ✔ | | | | ✔ | |
| [164] | 2018 | | ✔ | | | | | | | | | | | | | | ✔ | ✔ | | | ✔ | |
| [165] | 2006 | | ✔ | | | | | | | | | ✔ | | | | | ✔ | ✔ | ✔ | | | |
| [166] | 2012 | | ✔ | | | | | | | | | ✔ | | | | | ✔ | | | ✔ | ✔ | |
| [167] | 2012 | | ✔ | | | | | | | | | ✔ | | | | | ✔ | ✔ | | | ✔ | |
| [168] | 2015 | | ✔ | | | | | | | | | ✔ | | | | | ✔ | ✔ | ✔ | | | |
| [169] | 2011 | | ✔ | | | | | | | | | | | | | | ✔ | | | | | ✔ |
| [170] | 2016 | | ✔ | | | | | | | | | ✔ | | | | | ✔ | | ✔ | | | |
| [171] | 2007 | | ✔ | | | | | | | | | ✔ | | | | | ✔ | ✔ | | | | |
| [172] | 2017 | | ✔ | | | | | | | | | ✔ | | | | | ✔ | ✔ | ✔ | | ✔ | |
| [165] | 2006 | | ✔ | | | | | | | | | ✔ | | | | | ✔ | ✔ | ✔ | | | |
| [173] | 2018 | | ✔ | | | | | | | | | | | | | | ✔ | ✔ | | | | |
| [174] | 2014 | | ✔ | | | | | | | | | ✔ | | | | | ✔ | ✔ | ✔ | | | |
| [175] | 2007 | | ✔ | | | | | | | | | ✔ | | | | | ✔ | ✔ | ✔ | | | |
| [176] | 2019 | | ✔ | | | | | | | | | ✔ | | | | | ✔ | ✔ | ✔ | | | |
| [177] | 2014 | | ✔ | | | | | | | | | ✔ | | | | | ✔ | ✔ | ✔ | | | |
| [178] | 2017 | | ✔ | | | | | | | | | ✔ | | | | | ✔ | ✔ | ✔ | | | |
| [179] | 2021 | | ✔ | | | | | | | | | ✔ | | | | | ✔ | ✔ | ✔ | | | |
| [180] | 2016 | | | ✔ | | | | | | | | ✔ | | ✔ | ✔ | | | ✔ | | | | ✔ |
| [181] | 2009 | | | | ✔ | | | | | | | ✔ | | ✔ | | ✔ | | ✔ | | | | |
| [182] | 2017 | | | ✔ | | | | | | | | ✔ | | ✔ | ✔ | | | ✔ | ✔ | | | |
| [183] | 2013 | | | | ✔ | | | | | | | ✔ | | ✔ | | ✔ | | ✔ | | | | |
| [184] | 2000 | | | | ✔ | | | | | | | ✔ | ✔ | ✔ | | ✔ | | ✔ | ✔ | | | |
| [185] | 2008 | | | ✔ | | | | | | | | ✔ | | ✔ | ✔ | | | ✔ | ✔ | | | |
| [186] | 2014 | | | | ✔ | | | | | | | | | | | ✔ | | ✔ | ✔ | | | |
| [187] | 2002 | | | | ✔ | | | | | | | ✔ | | ✔ | | ✔ | | ✔ | ✔ | | | |
| [188] | 2007 | | | | ✔ | | | | | | | ✔ | | ✔ | | ✔ | | ✔ | ✔ | | | |
| [120] | 2015 | | | | ✔ | | | | | | | ✔ | | ✔ | | ✔ | | ✔ | ✔ | | | |
| [121] | 2003 | | | | ✔ | | | | | | | ✔ | | ✔ | | ✔ | | ✔ | ✔ | | | |
| [145] | 2004 | | | | ✔ | | | | | | | ✔ | | ✔ | | ✔ | | ✔ | | ✔ | | |
| [189] | 2018 | | | | | | | | | | ✔ | | | | | ✔ | | ✔ | ✔ | | | |
| [190] | 2001 | | | | | ✔ | | | | | | | | | ✔ | | | ✔ | | | ✔ | |
| [191] | 2002 | | | | | ✔ | | | | | | ✔ | | ✔ | ✔ | | | ✔ | ✔ | | | |
| [192] | 2008 | | | | | ✔ | | | | | | ✔ | | ✔ | ✔ | | | ✔ | ✔ | | | |
| [193] | 2008 | | | | | ✔ | | | | | | ✔ | | ✔ | ✔ | | | ✔ | ✔ | | ✔ | |

**Table 2.** *Cont.*

| Study | Year | Method | | | | | | | | | | | | Structure | | | | Evaluated Parameter | | | | |
|---|---|---|---|---|---|---|---|---|---|---|---|---|---|---|---|---|---|---|---|---|---|---|
| | | a | b | c | d | e | f | g | h | i | j | k | l | Column | Frame | Wall | Bridge Piers | SR | ED | SO | St | DR |
| [194] | 2005 | | | | | ✔ | | | | | | ✔ | | ✔ | | | | ✔ | ✔ | | | |
| [195] | 2010 | | | | | ✔ | | | | | | ✔ | | ✔ | | | | ✔ | | | | |
| [196] | 2010 | | | | | ✔ | | | | | | ✔ | | ✔ | | | | ✔ | ✔ | | | |
| [197] | 2014 | | | | | ✔ | | | | | | ✔ | | ✔ | | | | ✔ | ✔ | | | |
| [198] | 2014 | | | | | ✔ | | | | | | ✔ | | ✔ | | | | ✔ | | | | |
| [199] | 2013 | | | | | ✔ | | | | | | ✔ | | ✔ | | | | ✔ | ✔ | | | |
| [200] | 2013 | | | | | ✔ | | | | | | ✔ | | ✔ | | | | ✔ | ✔ | | | |
| [201] | 2018 | | | | | ✔ | | | | | | ✔ | | ✔ | | | | ✔ | | | | |
| [202] | 2017 | | | | | ✔ | | | | | | ✔ | | ✔ | | | | ✔ | | | | |
| [203] | 2018 | | | | | ✔ | | | | | | | | ✔ | | | | ✔ | ✔ | | | |
| [204] | 2021 | | | | | ✔ | | | | | | | | ✔ | | | | ✔ | ✔ | | | |
| [205] | 2008 | | | | | | ✔ | | | | | ✔ | | ✔ | | | | ✔ | ✔ | | | |
| [206] | 2016 | | | | | | ✔ | | | | | ✔ | | ✔ | | | | ✔ | ✔ | | | |
| [207] | 2017 | | | | | | ✔ | | | | | ✔ | | ✔ | | | | ✔ | ✔ | | | ✔ |
| [208] | 2020 | | | | | | ✔ | | | | | ✔ | | ✔ | | | | ✔ | ✔ | | | |
| [209] | 2017 | | | | | | | ✔ | | | | ✔ | | | | ✔ | | ✔ | | | | |
| [210] | 2018 | | | | | | | ✔ | | | | ✔ | | | | ✔ | | ✔ | ✔ | | | |
| [211] | 2016 | | | | | | | ✔ | | | | ✔ | | | | ✔ | | ✔ | ✔ | | | |
| [212] | 2019 | | | | | | | ✔ | | | | ✔ | | | | ✔ | | ✔ | ✔ | | | |
| [213] | 2014 | | | | | | | ✔ | | | | | | | | ✔ | | ✔ | ✔ | | | |
| [214] | 2018 | | | | | | | ✔ | | | | | | | | ✔ | | ✔ | ✔ | | | |
| [215] | 2020 | | | | | ✔ | | ✔ | | | | ✔ | | | ✔ | ✔ | | ✔ | ✔ | | | |
| [216] | 2021 | | | | | ✔ | | ✔ | | | | ✔ | | | ✔ | ✔ | | ✔ | ✔ | | | |
| [217] | 2001 | | | | | | | | ✔ | | | ✔ | | | | ✔ | | ✔ | ✔ | | | ✔ |
| [218] | 2004 | | | | | | | | ✔ | | | ✔ | | | | ✔ | | ✔ | | | | |
| [219] | 2006 | | | | | | | | ✔ | | | ✔ | | | | ✔ | | ✔ | ✔ | | | |
| [220] | 2016 | | | | | | | | ✔ | | | ✔ | | | | ✔ | | ✔ | | | | |
| [221] | 2017 | | | | | | | | ✔ | | | ✔ | | | | ✔ | | ✔ | ✔ | | | ✔ |
| [222] | 2009 | | | | | | | | ✔ | | | | | | | ✔ | | ✔ | ✔ | | | |
| [223] | 2019 | | | | | | | | | | ✔ | | | | | ✔ | | ✔ | ✔ | | | |

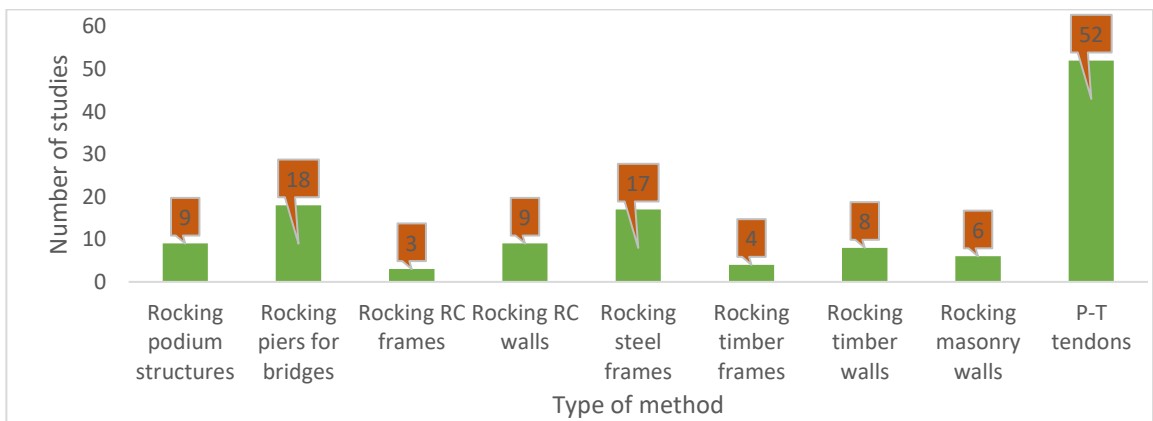

**Figure 10.** Visualization of the summarized research based on the method discussed.

## 5. Research Needs

This paper has presented a summary of the state of the art of the research dealing with problems related to the stability, strengthening, and resilience of RC structures. It clearly revealed the urgent need to develop a feasible, efficient, and applicable technique to

reduce residual displacement and enhance resilience in shear walls. In the last few decades, extensive research has been devoted to FRPs and their applications for strengthening RC structures. In parallel, many researchers have evaluated self-centering improvement methods and residual displacement reduction techniques in reinforced concrete structures, specifically, shear walls. As shown in Figure 10, most of the research carried out so far has been related to either PT and rocking methods or systems using mechanical devices, which can only be used in the design of new structures. Considering the excellent performance of EB-FRP in the seismic strengthening of concrete structures and the need to strengthen existing shear walls against large deformations, there is a need to conduct comprehensive research on the use of EB-FRP to increase the self-centering ability of existing RC structures. The exact expression of the required configuration, width, and number of FRP sheets, the mechanical characteristics, and the way to connect FRP to concrete are among the issues that need to be addressed in this research. Some other significant concerns that should be studied are as follows: (i) evaluate the changes in parameters after the use of EB-FRP in SSWs and CSWs, such as displacement and residual displacement, hysteretic behavior, displacement ductility, failure mode, sequence of failure, stiffness degradation, energy dissipation, strength retention, and sequence of hinge formation; (ii) optimize EB-FRP properties and configuration for CSWs and SSWs to achieve the lowest residual displacement. Figure 11 illustrates the parameters that could influence the residual displacement in strengthened shear walls using EB-FRP.

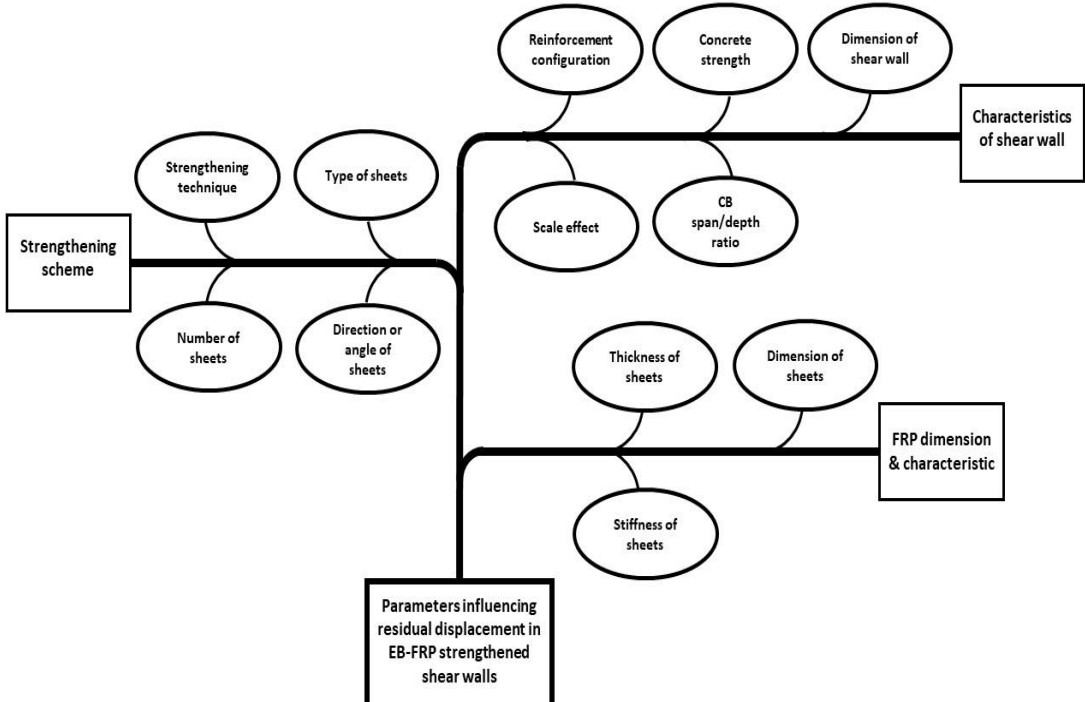

**Figure 11.** Overview of the parameters influencing residual displacement in EB-FRP strengthened shear walls.

## 6. Conclusions

This paper presents the state of the art of self-centering enhancement methods in shear walls and FRP applications for strengthening RC structures. Evaluating their advantages and disadvantages and considering the importance of paying attention to issues such as failure mode and sequence of failure, ductility, energy dissipation, and sequence of hinge formation, the following conclusions were drawn:

- RC structures strengthened with EB-FRP sheets have shown outstanding performance in seismic events. This reliable technique can improve RC structures' shear, flexural,



axial, fatigue, and impact strength using different configurations such as side bonding, U-wrap, and complete wrapping.

- Most studies have been done to improve the self-centering ability in new structures, mainly using PT tendons, SMAs, and other innovative devices. Few studies have focused on self-centering enhancement in existing structures.
- Currently, there are no explicit data on the ability of EB-FRP to enhance the resilience and self-centering ability of existing shear walls; hence, future studies should focus more on evaluating this method's effectiveness.

Accordingly, the authors established a research need to develop methods and techniques to improve the self-centering ability of CSWs and SSWs using EB-FRP. In addition to being cost-effective, these methods will help the environment by preventing the deconstruction and replacement of concrete elements and thereby reducing $CO_2$ emissions in the environment resulting from cement production. The cement industry is one of the leading causes of global warming, with a share of cement production in global $CO_2$ emissions of approximately 7%.

**Author Contributions:** Conceptualization, A.A. and O.C.; methodology, A.A. and O.C.; Literature review, A.A.; investigation, A.A. and O.C.; resources, O.C.; writing—original draft preparation, A.A.; writing—review and editing, O.C.; supervision, O.C.; project administration, O.C.; funding acquisition, O.C. All authors have read and agreed to the published version of the manuscript.

**Funding:** O.C. is funded by the National Science and Engineering Council (NERC) of Canada and by the Fonds de Recheche du Quebec-Nature & Technologie (FRQ-NT).

**Data Availability Statement:** The data supporting the finding of this study are available within the article.

**Acknowledgments:** The financial support of the Natural Sciences and Engineering Research Council of Canada (NSERC) and the Fonds de recherche du Québec–Nature et technologie (FRQNT) through operating grants is gratefully acknowledged.

**Conflicts of Interest:** The authors declare no conflict of interest.

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
