# Peer review of "Enhancing Resilience and Self-Centering of Existing RC Coupled and Single Shear Walls Using EB-FRP: State-of-the-Art Review and Research Needs"

_jcs, doi:10.3390/jcs6100301_

Round 1

Reviewer 1 Report

In the title of the article, the authors suggest the use of EB-FRP for self-centering of existing RC coupled and single shear walls. However, at the beginning of the manuscript they describe concrete elements reinforced with steel. This requires clarification and ordering in the context of the use of FRP, which, due to the material characteristics, is significantly different from steel reinforcement.

The article is not suitable for publication in JCS as it stands.

General remarks

1. The summary should also include information on remedial measures enhancing the resistance of RC structures to seismic forces.

2. In the introduction, the authors briefly informed about the advantages and disadvantages of FRP composites. The specific advantages and disadvantages of FRP should be identified.

3. In table 1; how was k, a, b, c determined? Why are they dependent on the number of storeys?

4. In subsection 2.1.2, the authors first report on the adhesion behavior of steel and concrete, and then go to FRP without explanation. Please explain the difference in adhesion to concrete between FRP bars and steel bars.

5. It is not clear whether the Authors in subsection 2.1 discuss the behavior of CCA with steel or composite reinforcement. Please make this clear in the article.

6. Chapter 3 discusses the benefits of FRF, but makes no mention of insects. Please write in the article also about the disadvantages.

7. Conclusions should be bulleted, specific and relate to the content of the manuscript.

The article is not well written and needs improvement.

Specific remarks

8. 43 line: Incomprehensible. Please formulate your sentence differently.

9. 91 line: It is not clear whether this applies to the CB or the entire CSW system. Please explain.

10. 96 line: Please correctly define P and lcg in the article.

11. 102 line: it should be added that k, a, b, c are dimensionless values. How were they established? Why are they dependent on the number of storeys?

12. 105 line, Tab.1: It should be better: "Number of storeys" not "Number of stories"

13. 107 line: Be clear about ɤ, ɤ? and ɤ?. In addition, it is not specified what Ie is.

14. 160 line: In modern, i.e. what standards? Please list them.

15. 230 line: Incomprehensible. Please formulate your sentence differently.

16. 309 line: In what units is the displacement ductility factor (??)?

17. 312 line: Incomprehensible. Please formulate the sentence correctly and explain the quantities.

18. 317 line: Incomprehensible. Please provide the definition of the parameter (factor).

19. 368 line: Please explain the differences between yield displacement and plastic displacement.

20. 382 line: What does Af mean? Please describe.

21. 397 line: What do the symbols ∅?, ∅? mean in equations 7-9?

22. 608 line; What does it mean in equation (10): ????, Ts, ?????

23. 627 line: Please explain in the article what is hybrid FRP reinforcement.

24. 736 line: Please write what DC stands for.

25. 781, 784 lines, Fig. 11, 12: What do the values ​​on the vertical axis mean?

 I recommend an in-depth review of the manuscript, including comments, to make it an article suitable for publication in the JCS.

Reviewer 2 Report

The reviewer recommends rejecting this work. The main reasons for the rejection are the following:

-       The study provides no scientific rationale, which causes the illogical structure of the presentation. It composes numerous repetitions and irrelevant considerations. For example, the first two subsections of Section 3.1 do not relate to the considered topic. In addition, the first two sections of the manuscript describe a superficial and fragmentary view of the problem; Figure 1 is primitive. On the other hand, the relevant literature descriptions lack essential details and conclusions. The following wording describes the typical example: “Christidis et al. [51] evaluated the flexural and shear deformation in shear walls through numerical and experimental analysis.” And what is the conclusion?

-       Wrong research methodology. The bibliography is non-qualified; the Authors have collected 290 (!) literature sources without any system. Often, the Authors formulate statements without reference to literature, e.g., “The primary deficiencies related to SSWs and CSWs can be summarized as follows….” Who has proposed such a classification? Moreover, as mentioned in the previous comment, a substantial part of the discussed literature problems lay beyond this investigation scope. First, the Authors had to introduce the research object (i.e., the shear wall with EB-FRP reinforcement system) and describe the relevant mechanical resistance mechanisms. Section 4 could form the review core, providing the development insights for the self-centering wall systems. Unfortunately, the superficial analysis did not allow the Authors to make tangible insights. For example, the fundamental Table 2 appears in the text without any discussion, which is unacceptable for scientific works. On the contrary, the literature review had to clarify a problem and, if possible, describe the solution. Unfortunately, this submission includes just a list of articles accompanied by minimal information; the Authors do not formulate/substantiate any concepts essential for engineering practice, ensuring additional value to the summarized information regarding the already published works.

-       The description absence of fundamental principles and approaches describes another critical drawback. It is well known that the analysis outcomes are closely related to the assumed simplifications and material models. However, the manuscript clarifies neither theoretical concepts nor modeling approaches suitable for developing self-centering structures with EB-FRC systems. Thus, it is impossible to identify the improving ways of the theoretical models (e.g., equations (4)–(9)).

-       Conclusions are uninformative and rudimentary in consequence of the drawbacks mentioned above.

-       The writing style is below any acceptance limits. The formulations like “Many (?) existing RC shear walls…”, “This (?) demonstrates the keen interest…”, and “Many (?) studies were performed…” are uninformative and irrelevant for scientific publications. The Reviewer found numerous grammatical errors, typos, and stylistically imperfect wordings in the manuscript. In addition, the Authors make no difference between beams, columns, and shear walls. This report does not include a list of the errors because of the excessive number of such occurrences.

Reviewer 3 Report

     The article provides an interesting overview. The reviewer recommends taking the following notes into account.

The objectives stated in the summary and introduction are not the same.

It is stated that the stiffness core is only needed to withstand lateral loads, but since you also mention seismic, then it should be also mentioned the force generated by mass.

The first figure should specifically indicate the research object and factors.

For the first paragraph of the second chapter, it would be good to present a scheme or picture so that the reader can understand the calculation scheme, loads and internal forces, resistance of major elements?

Revise the second equation, where is the moment of inertia, for which walls the third equation applies?

Whether the beams must withstand transverse forces or simply the cores should cause smaller transverse strains, because it is not wise to throw everything to the beams?

It is not clear what is depicted in the fourth picture, is it a cross-section or what? Where are the internal forces?

Please explain equation 4 and 5, what do they calculate?

C – neutral axis distance (equation 6). From where this distance?

Please explain formulas 7; 8; 9 and their members.

“In another study by Honarparast et al. [173], two… and CSA A23.3 14. [29] Coupling beams’… The study clearly revealed…” Where does the sentence end, which study? Hard to understand.

Explain tfrp in equation 10 and 11.

Explain the members of the 12 equation sequentially.

How to understand: elsewhere [178] and by Elnady [179]. M. A. Sakr and others in elsewhere?

PT elements or P-T? And PT explanation is given later.

Figure 8 mentioned earlier than 7.

“deformations. however” in line 721

Please explain vertical axe in pictures 11 and 12.

Conclusions could be made in relation to the objectives set.

Round 2

Reviewer 1 Report

Compared to the previous version of the article, they have introduced corrections that partially reflect the reviewer's suggestions. However, there are still a few inaccuracies that need to be clarified.

1. There is still no answer to the second question in note 3.

2. Note 4 on bond to concrete of steel and FRP was not answered.

3. There was no reply to Observation 14 on "Modern Standards".

The article may be published after considering the above comments.

Reviewer 2 Report

This work provides the reader with no additional information regarding already published works. Such a drawback is unacceptable for scientific articles. The Authors did not improve the writing style and the presentation structure:

  • Section 1 is uninformative and unnecessary.
  • Section 2 presents a superficial and primitive view of the problem. For instance, Figure 3 is wrong (considering the distributed load effect); it appears in the text without any comments. The newly introduced text (Lines 96–165) repeats existing formulas without describing the resisting mechanisms causing the considered effects, etc.
  • Section 3 is too general – it can regard the strengthening of any concrete structure. Moreover, it appears in the text without explaining the necessity of strengthening the self-centering shear wall systems. What are the problems which these structures face? Again, this section only repeats the already published results. What is the Authors’ opinion about the design tendencies and perspectives? Figure 5 appears in the text without any comments. This drawback is unacceptable for scientific publications.
  • Section 4 could form the core of the review article. Still, the Authors do not understand the principles of such research – such articles must extend the current knowledge by presenting the general view of the problem, formulating original ideas, and highlighting the research problems and trends. In addition, Figures 6 and 8 are far too primitive; Figure 9 appears in the text without any comments; a very primitive comment accompanies Table 2, which could represent a relevant object for discussions, etc.
  • Section 5 is far too primitive.
  • Conclusions are irrelevant because of the drawbacks described above.

Reviewer 3 Report

The reviewer does not like the transverse force distribution diagram shown in the second figure, as well as the title of the figure.

Explanations provided by the authors, point 13 and point 14 are not finally resolved.

The authors present many equations, but do not explain all their members.

Round 3

Reviewer 1 Report

The response to the difference in adhesion of FRP bars and steel bars to concrete shows that the authors do not want to provide information on this subject.

Reviewer 3 Report

The title of the second picture does not correspond to the idea presented in the text. Transverse force in the figure name and lateral force in the text. Shear diagram is common for the frame not a wall structural element. The authors could use a finite element program for verification.

Authors still use the expression: elsewhere, for citation. Changed it in one place and left it in another.

Authors present information with errors and do not correct, so the reviewer does not have a positive recommendation for publication.

Round 4

Reviewer 3 Report

Good luck